# Oleanolic Acid Nanofibers Attenuated Particulate Matter-Induced Oxidative Stress in Keratinocytes

**DOI:** 10.3390/antiox10091411

**Published:** 2021-09-02

**Authors:** Hsuan Fu, Feng-Lin Yen, Pao-Hsien Huang, Chun-Yin Yang, Chia-Hung Yen

**Affiliations:** 1Department of Fragrance and Cosmetic Science, College of Pharmacy, Kaohsiung Medical University, Kaohsiung 807, Taiwan; susan070287f@gmail.com (H.F.); flyen@kmu.edu.tw (F.-L.Y.); j6466497@gmail.com (P.-H.H.); anitayang0812@gmail.com (C.-Y.Y.); 2Master Degree Program in Toxicology, College of Pharmacy, Kaohsiung Medical University, Kaohsiung 807, Taiwan; 3Drug Development and Value Creation Research Center, Kaohsiung Medical University, Kaohsiung 807, Taiwan; 4Department of Medical Research, Kaohsiung Medical University Hospital, Kaohsiung 807, Taiwan; 5School of Pharmacy, College of Pharmacy, Kaohsiung Medical University, Kaohsiung 807, Taiwan; 6Institute of Biomedical Sciences, National Sun Yat-Sen University, Kaohsiung 804, Taiwan; 7Graduate Institute of Natural Products, College of Pharmacy, Kaohsiung Medical University, Kaohsiung 807, Taiwan; 8National Natural Product Libraries and High-Throughput Screening Core Facility, Kaohsiung Medical University, Kaohsiung 807, Taiwan

**Keywords:** particulate matters, oleanolic acid, nanofiber, antioxidant, anti-inflammatory, anti-aging

## Abstract

Airborne particulate matter (PM) is one of the indicators of air pollution, and it is also the main factor causing oxidative stress in the skin. Oleanolic acid (OA), a natural terpenoid compound, effectively inhibited PM-induced skin aging; however, OA has poor water solubility and skin absorption, which limit its application in medicines and cosmetics. The aim of this study was to prepare oleanolic acid nanofibers (OAnf) and evaluate the effects of OA and OAnf in PM-treated keratinocytes. The results showed that OA dissolved in dissolved in dimethyl sulfoxide (DMSO) attenuated PM-induced reactive oxygen species overproduction, stress-activated protein kinase/Jun-amino-terminal kinase (SAPK/JNK) activation, and the expressions of inflammatory and skin-aging-related proteins. In addition, the nanofiber process of OA effectively improved the water solubility of OA more than 99,000-fold through changing its physicochemical properties, including a surface area increase, particle size reduction, amorphous transformation, and hydrogen bonding formation with excipients. The skin penetration ability of OAnf was consistently over 10-fold higher than that of OA. Moreover, when dissolved in PBS, OAnf displayed superior antioxidant, anti-inflammatory, and anti-skin aging activities in PM-treated keratinocytes than OA. In conclusion, our findings suggest that OAnf could be a topical antioxidant formulation to attenuate skin problems caused by PM.

## 1. Introduction

Air pollution is now a public health problem across the whole world. With the development of technology, various harmful substances, including gases, chemical substances, biological pollutants, and particles, accumulate in the atmosphere and severely affect people’s lives and health. Particulate matter (PM), an indicator of air pollutants, is a combination of various organic compounds, biologically derived materials, and particulate carbon nuclei [1]. PM enters the lungs by inhalation and enters into the blood circulation causing systemic health hazards such as organ inflammation and cardiovascular and respiratory diseases [2,3]. In addition, PM can pass through the skin barrier and accumulate in the hair follicles and even penetrate into the dermis with repeated contact; therefore, overexposure of the skin to PM has been associated with extrinsic skin aging, changes in pigmentation, atopic dermatitis, acne, and psoriasis [2,4]. Prolonged contact with PM induces the overproduction of reactive oxygen species (ROS) in keratinocytes, which triggers several signaling pathways including the apoptosis pathway, mitogen-activated protein kinase (MAPKs) pathways, and inflammation. Elevated expression of cyclooxygenase-2 (COX-2), tumor necrosis factor-α (TNF-α), and interleukin-1β (IL-1β) is commonly seen in keratinocytes exposed to PM. Moreover, PM also activates matrix metalloproteinases (MMPs) and results in the loss of skin elasticity and aging [5].

Natural products have long been used as effective cosmeceutical ingredients. Oleanolic acid (OA, 3β-hydroxyolean-12-en-28-oicacid), a five-ring triterpenoid compound, is widely present in plants, fruits, and vegetables [6]. OA is well known for its hepatic protective effects such as reducing chemical-induced acute liver damage and fibrosis/cirrhosis in chronic liver diseases [6,7]. In addition, previous studies have revealed that OA possesses antioxidant, anti-cancer, anti-inflammatory, anti-diabetics, anti-microbial effects [8]. Kim et al. revealed that OA could decrease the pro-inflammatory cytokine (TNF-α, IL-6) and skin aging protein (MMP-1) expression in PM-treated keratinocytes [9]. However, the physicochemical properties of OA make it difficult to dissolve in water, which limits its application in medicines, food, and cosmetics.

The formulation designs of drug delivery such as polymer-based nanocarriers, liposomes, and nanofibers are usually used to improve physicochemical properties of the active ingredients. Encapsulation of active ingredients with excipients in these pharmaceutical formulations could enhance their water solubility and skin absorption and lower potential toxicity and irritation to the skin. Among them, nanofibers are an emerging nanosized formulation, with a large surface area, low density, and a high pore volume, and are already widely used in biomedicine, which can reduce the volume of oral drugs, increase the stability of active ingredients, control release, improve bioavailability, and make artificial tissues [10]. Electrospinning is a common technology used to produce nanofibers and is highly compatible with mass production [11]. Therefore, the preparation of nanofiber using the electrospinning process can simultaneously improve the bioavailability and production efficiency of an active ingredient with poor water solubility. Polyvinyl pyrrolidone (PVPK90) and 2-hydroxypropyl-β-cyclodextrin (HPBCD) are FDA-approved compounds for solubilizing and delivering hydrophobic active pharmaceutical ingredients in humans. Previous studies showed that nanofibers prepared with HPBCD and PVPK90 significantly improved the water solubility and skin penetration of resveratrol [12] and plai oil [13]. Thus, the aim of this study was to use PVPK90 and HPBCD as delivery carriers to prepare the oleanolic acid nanofibers (OAnf) and to evaluate the effects of OA and OAnf in PM-treated keratinocytes.

The aim of the present study was to evaluate the biological effect of OA dissolved in DMSO in PM-induced keratinocytes damage. To overcome the poor water solubility of OA, we used PVPK90 and HPBCD as delivery carriers to prepare oleanolic acid nanofibers (OAnf) by an electrospinning process and then determined the changes of physicochemical properties between raw OA and OAnf to elucidate the improvement of water solubility and skin penetration. To compare the biological effect after the nanofiber process of OA, a PM-induced keratinocytes damage model was used to evaluate the antioxidative, anti-inflammatory, and anti-aging activity of OAnf and OA.

## 2. Materials and Methods

### 2.1. Materials

Oleanolic acid hydrate (OA) was purchased from Tokyo Chemical Industry Co., Ltd. (Tokyo, Japan). Polyvinyl pyrrolidone (Luviskol^®^ K90 Powder, PVP) was purchased from Wei Ming Pharmaceutical Mfg. Co., Ltd. Taipei, Taiwan). Hydroxypropyl-beta-cyclodextrin (HPBCD) was obtained from Zibo Qianhui (Zibo, China). Methanol and dimethyl sulfoxide (DMSO) were purchased from Aencore Chemical (Surrey Hills, Australia). All chemicals or reagents for cell cultures were biological grade, and other chemicals of physicochemical determination were of high-performance liquid chromatography (HPLC) grade.

### 2.2. Cell Viability Assay

The cell viability determination of an active ingredient is a common method used to choose the proper concentration ranges for active ingredients to evaluate their biological activity. HaCaT keratinocytes were purchased from Istituto Zooprofilattico Sperimentale della Lombardia edell’Emilia Romagna (Brescia, Italy). HaCaT cells were cultured in DMEM (Himedia Laboratories, Mumbai, India) containing 10% fetal bovine serum (Hazelton Product, Denver, PA, USA) with 1% penicillin–streptomycin (Biological Industries, Connecticut, NE, USA), and HaCaT cells were incubated in an incubator (Thermo Fisher Scientific, Waltham, MA, USA) with the conditions set at 37 °C with 5% CO_2_. For preparation of the test samples, OA and OAnf were dissolved in DMSO and PBS, respectively, and then each sample was diluted in DMEM without fetal bovine serum for cell viability determination. The HaCaT cells were seeded in 96-well plates at a density of 1 × 10^4^ cells/100 μL/well for 24 h. The culture medium was then removed, and the cells were treated with different concentrations of OA and OAnf ranging from 5 to 80 μM in serum-free DMEM for 24 h. At the time of assay, the treatment medium was removed, and 150 μL of 0.5 mg/mL MTT solution was added into each well. After 3 h of incubation, the MTT solution was removed, and the purple formazan crystals of each well was dissolved in 100 μL of DMSO. The absorbance at 550 nm of each well was then measured using a microplate spectro-photometer (BioTek μQuant, Winooski, VT, USA). The cell viability was calculated by the following formula:(1)$$\mathrm{Cell}\;\mathrm{viability}\;(\%)=\frac{{\mathrm{OD}}_{\mathrm{sample}}}{{\mathrm{OD}}_{\mathrm{control}}}\times 100\%$$

### 2.3. Determination of Reactive Oxygen Species (ROS) Content

PM (Standard Reference Material, SRM^®^ 1649b) were purchased from the National Institute of Standards and Technology. This product was collected in 1976 and 1977 in Washington, DC Made. A total of 10 mg/mL of PM were suspended in PBS and then sonicated for 10 min before use. A total of 1 × 10^4^ HaCaT keratinocytes were cultured in 96-well plates for 24 h under 37 °C and 5% CO_2_ conditions. Cells were treated with different concentrations of OA in DMSO, OA in PBS, and OAnf in PBS for 24 h, respectively. Then, they were incubated with 20 μM dichlorodihydrofluorescein diacetate (DCFH-DA; Sigma, Tokyo, Japan) solution for 30 min. Next, 50 μg/cm^2^ PM was added into each well and incubated for 1 h. After that, cells were washed twice with PBS, and the fluorescence intensity of each sample was analyzed using the fluorescent plate reader (excitation: 485 nm; emission: 528 nm) (BioTek, Winooski, VT, USA). The following equation was used to calculate the inhibition percentage of ROS production:(2)$$\mathrm{ROS}\;\mathrm{production}\;(\%)=\frac{{\mathrm{OD}}_{\mathrm{sample}}}{{\mathrm{OD}}_{\mathrm{control}}}\times 100\%$$

### 2.4. Western Blot Analysis

A total of 4 × 10^5^ HaCaT keratinocytes were cultured in 6-well plates for 24 h. Cells were then treated with OA or OAnf in serum-free medium for 24 h followed by the addition of PM. After various time points, cells were lysed with RIPA lysis buffer (Merck Millipore, Burlington, MA, USA), then centrifuged at 12,000 rpm for 10 min. A BCA protein assay kit (Thermo Fisher Scientific, Waltham, MA, USA) was used to determine protein concentration. Then, the proteins were separated by sodium dodecylsulfate–polyacrylamide gel electrophoresis (SDS-PAGE), and then blotted onto polyvinylidene difluoride (PVDF) membranes (Merck Millipore). Membranes were blocked for 1 h and washed with Tris-buffered saline (TBS) with 1% Tween-20. Membranes were incubated with primary antibodies at 4 °C overnight. The primary antibodies used in this study included cyclooxygenase-2 (COX-2), matrix metalloproteinase-9 (MMP-9), tissue inhibitor of metallopro-teinase-1(TIMP-1), stress-activated protein kinase/Jun-amino-terminal kinase (SAPK/JNK) (Cell Signaling Technology, Danvers, MA, USA), GAPDH (Santa Cruz Bi-otechnology, Dallas, TX, USA), matrix metalloproteinase-1 (MMP-1) (Proteintech Group, Rosemont, IL, USA), p38α, extracellular regulated protein kinases (ERK), and nuclear factor kappa-light-chain-enhancer of activated B cells (NF-κB) (Merck Millipore, Burlington, MA, USA). Next, secondary antibodies were added for 1 h at room temperature and reacted with enhanced chemiluminescence reagents (ECL; Thermo Fisher Scientific). Antibodies against GAPDH were used as internal controls. Each protein’s expression was analyzed using a Touch Imager (e-BLOT; Shanghai, China), and the expression was quantified using ImageJ.

### 2.5. Preparation of Oleanolic Acid Nanofibers (OAnf)

OAnfs were electrospun with different ratios of OA:PVP:HBPCD (1:8:5, 1:8:10, and 1:8:20). The electrospun solution was prepared as follows: 25 mg of OA was dissolved in 5 mL of methanol, and HPBCD was added and stirred with a magnet stirrer to obtain a clear solution; then, PVPK90 was immediately added, and the mixture was stirred for 1 h. The nanofibers were woven using FES-COS Electro-spinning equipment (Falco Tech Enterprise Co., Taipei, Taiwan) under the following conditions: a 10 mL syringe with a needle of internal diameter of 0.22 mm was employed for electrospinning; the flow rate was adjusted to 0.2 mL/h; the applied voltage was set at 12 KV; the tip-collector distance was 10 cm. After the electrospinning process, the nanofibers were collected using aluminum foil. The newly synthesized nanofibers were placed in a sealed plastic bag and stored in a moisture-proof container.

### 2.6. High-Performance Liquid Chromatography (HPLC) Analysis of Oleanolic Acid

The HPLC analysis system (LaChrom Elite L-2000, Hitachi, Tokyo, Japan) consisted of an L-2130 pump, an L-2200 autosampler, and an L-2420 ultraviolet-visible (UV-vis) detector. The analysis column was a Mightysil RP-18 GP column (250 × 4.6 mm i.d., 5 μm). The mobile phase was composed of methanol and 0.1% glacial acetic acid solution in a fixed ratio (95:5; *v*/*v*). The flow rate of the mobile phase was 1 mL/min, and the detection wavelength of the UV detector was set at 215 nm. The absorption peak for oleanolic acid appeared at 7.5 min. The calibration curve of oleanolic acid displayed a good linear (*r* = 0.999) within the range 0.01–100 μg/mL.

### 2.7. Morphology, Fiber Diameter, and Particle Size Measurement of OAnf

Different samples of nanofibers were plated with platinum with an ion coater (E-1045, HITACH, Tokyo, Japan); the condition was set at 10 mA 120 s later. The morphology and shape of each sample was observed by a scanning electron microscope (Hitachi S4700, Hitachi, Tokyo, Japan). The diameter of each sample was calculated by image j software. A Zetasizer 3000HS analyzer (Malvern, Worcestershire, UK) was used to measure the particle size of OAnf. The particle size of OA and OAnf were measured at a concentration of 1 mg/mL and 0.1 mg/mL, respectively. In addition, we also observed the uniformity of the morphology of OAnf after dissolving in water using a transmission electron microscope (TEM, JEM-2000EXII instrument, JEOL Co., Tokyo, Japan). The test sample was adjusted to 1 µg/mL of OA in deionized water and then dripped into the copper mesh, and then 0.5% (*w*/*v*) phosphotungstic acid was immediately dripped. After drying, each sample was placed on the TEM for observation.

### 2.8. Drug Loading and Encapsulation Efficiency of OAnf

It is highly important to determine the drug loading and encapsulation efficiency of the delivery system for evaluating the performance of the pharmaceutical process. The drug loading was calculated as the percentage of the determined content and the theoretical content of the OA contained in the OA nanofibers. For drug loading determination, 100 μL of each sample was added into 900 μL of methanol, and the OA concentration was immediately measured by the aforementioned HPLC method. The following equation was used to calculate drug loading:(3)$$\mathrm{Drug}\;\mathrm{loading}\;\left(\%\right)=\frac{C_{\mathrm{OA}}\times {\;V}_{\mathrm{OAnf}}}{W_{\mathrm{OA}}}\times 100\%$$
where C_OA_ is the concentration of OA from OAnf, W_OA_ is the theoretical amount of OA added, V_OAnf_ is the volume of OAnf solution.

The encapsulation efficiency indicates whether the nanofibers successfully encapsulated the active compounds. OAnf samples was dissolved in deionized water, and was added into the centrifugal filter devices (Microcon YM-10, Millipore, Billerica, MA, USA), and then centrifuged at 12,000 rpm for 10 min by the refrigerated centrifuge (Centrifuge 5430R, Eppendorf, Hamburg, Germany). The encapsulated part was retained in the upper tube, and the unencapsulated part was collected from the lower tube due to the difference in molecular weight. The amount of unencapsulated OA was detected by the aforementioned HPLC method. The following equation was used to calculate encapsulation efficiency:(4)$$\mathrm{Encapsulation}\;\mathrm{efficiency}\left(\%\right)=\frac{A_{\mathrm{OA}}-A_{\mathrm{unentrapped}\;\mathrm{OA}}}{A_{\mathrm{OA}}}\times 100\%$$
where A_OA_ is the theoretical amount of OA (obtained from feeding condition) incorporated into the nanofibers, and A_unentrapped OA_ is the amount of unencapsulated OA.

### 2.9. Aqueous Solubility of OAnf

Raw OA (1 mg) and different ratio formulations of OAnf (containing an equivalent of 1 mg oleanolic acid) were dissolved in 1 mL deionized water, respectively, and then sonicated under a ultrasonicator (Branson 5510, Emerson Electric, St. Louis, MO, USA) for 20 min. Each sample was filtered through a 0.45 μm membrane (Pall Corporation, Washington, NY, USA), and diluted 10-fold. The diluted solutions were analyzed by HPLC, and the standard curve was employed to determine the oleanolic acid amount for compared their aqueous solubility.

### 2.10. Determination of Crystalline-to-Amorphous Transformation

X-ray diffractometry (Siemens D500, Karlsruhe, Germany) was used to analyze the crystalline form of OA, excipients and OAnf. The analysis was conducted using nickel-filtered Cu-Kα radiation, using a voltage of 40 kV and current of 25 mA. The scan rate was 1°/min, and the range of the angles scanned was from 5° to 50°.

### 2.11. Intermolecular Interaction between OA and Excipients

Fourier transform infrared spectroscopy (FTIR) and 1H Nuclear magnetic resonance (1H NMR) are usually used to confirm the intermolecular interaction between active ingredients and excipients. Oleanolic acid, PVP, HPBCD, and different ratio formulations of OAnf were, respectively, mixed with potassium bromide (KBr) in a volume ratio of 1:9 using a mortar and pressed into tablets. Each sample was then analyzed by the FTIR spectrophotometer (Perkin-Elmer 200 spectrophotometer, Perkin-Elmer, Norwalk, CT, USA). The scanning range was 400–4000 cm^−1^. In addition, each sample was dissolved in 0.8 mL of 99.8% DMSO-*d*_6_ (Merck, St. Louis, MO, USA) and analyzed by JEOL Alpha 400 spectrometer (Nihon Denshi Co., Tokyo, Japan).

### 2.12. Ex Vivo Skin Penetration of Oleanolic Acid and Its Nanofiber

This experiment was performed according to the European Cosmetic Toiletry and Perfumery Association (COLIPA) guidelines’ standard protocol. The Franz diffusion cell system can be divided into glass containers of the upper donor chamber and the lower receptor chamber. A total of 1.5 mL buffer solution comprising 0.14 M NaCl, 2 mM K_2_HPO_4_, 0.4 mM KH_2_PO_4_ (pH 7.4) was placed in the receptor chamber and stirred with a magnetic bar at 600 rpm throughout the experiment. Fresh flank skin from a pig was obtained from a local butcher in the market and refrigerated during the experiment period. Each skin sample was cut into 2 cm × 2 cm pieces and placed between the two chambers, with the *stratum corneum* facing upwards. The Franz diffusion cell was maintained at 32 °C with a circulating water bath. Then, 200 μL of 1 mg/mL of OA or OAnf was added to the donor chamber for 1, 2, or 4 h. After that, the pig skin was removed from the Franz diffusion cell, and the *stratum corneum* was obtained by tape stripping 15 times. Each residual skin sample was heated to 95 °C with a heat pad, and the epidermis and dermis were separated using a scalpel. Each sample was immersed in methanol and sonicated for 1 h to extract OA, and the content of oleanolic acid in each sample was determined by the HPLC method.

### 2.13. Statistical Analysis

All data were displayed as mean ± standard deviation (SD). The statistically significance between different groups were analyzed by analysis of variance (ANOVA) with Tukey’s post hoc test. *p* < 0.05 indicated statistical significance.

## 3. Results

### 3.1. Oleanolic Acid Can Suppress Inflammation, Aging, and ROS/MAPKs Signaling Pathways in PM-Induced Keratinocytes Damage

To find the proper concentration range for biological activity evaluation, the cytotoxicity of OA dissolved in DMSO was determined in human HaCaT keratinocytes cells using an MTT assay. As shown in Figure 1A, 40 and 80 μM of OA were associated with 32 to 16% cell viability. OA at less than 20 μM were still had a cell survival rate over 85%. These results indicated that OA at a concentration of 5–20 μM has no cytotoxic effects on human HaCaT keratinocytes (Figure 1A). Accordingly, OA was studied at a concentration of 5–20 μM to investigate its antioxidant and antipollution activity in PM-induced keratinocytes damage. Recently, many studies have demonstrated that PM is a common air pollutant causing ROS overproduction and subsequent damage of the skin system through a series of oxidative stress, including lipid peroxidation, protein carbonylation, and DNA mutation [14,15,16]. As shown in Figure 1B, PM treatment significantly increased the ROS production when compared with the untreated group (*p* < 0.05). In contrast, pretreatment with OA effectively decreased PM-induced ROS overproduction in a dose-dependent manner (*p* < 0.05). Therefore, these results suggested that OA possessed antioxidant activity to prevent PM-induced oxidative stress by reducing the ROS overproduction. In addition, ROS overproduction after PM exposure can activate the phosphorylation of MAPKs proteins, including p-ERK, p-p38, and p-JNK, triggering the protein expressions of inflammation and aging [17,18]. The present study also found that PM treatment can increase the expression of inflammatory proteins (COX-2 and NF-κB), skin aging-related proteins (MMP-1, MMP-9 and TIMP-1), and phosphorylation of ERK, JNK, and p38 (*p* < 0.05). Our present results also demonstrated that OA at 10 and 20 μM significantly inhibited the protein expression of NF-κB and COX-2 when compared with the PM treatment group (*p* < 0.05) (Figure 1C). Furthermore, OA pretreatment also effectively reversed PM-induced alteration on MMP-1 and TIMP-1 expression (*p* < 0.05) but had no effect on MMP-9 (Figure 1D). We further determined the effects of OA treatment on phosphorylation of MAPKs during PM exposure, and our results indicated that OA could inhibit the phosphorylation of JNK (*p* < 0.05), but had no effect on ERK or p38 (Figure 1E). According to the above results, when dissolved in DMSO, OA displayed good skin-protective activity and could ameliorate PM-induced ROS overproduction, JNK activation, and inflammatory and skin-aging protein expression in keratinocytes.

### 3.2. Oleanolic Acid Nanofibers Increased the Water Solubility and Skin Penetration of Raw Oleanolic Acid by Improving the Physicochemical Properties

#### 3.2.1. Surface Morphology of Oleanolic Acid and Its Nanofibers

Under SEM observation, the appearance of HPBCD presented a spherical and porous excipient, and its size was about 20 to 50 μm (Figure 2A). PVPK90 is an excipient with irregular polygonal particles and a particle size of more than 60 μm (Figure 2B). Raw oleanolic acid is an irregularly round granular powder with a size of 3–60 μm. Figure 2D–F show the mean fiber diameter of different weight ratios of oleanolic acid: HPBCD: PVPK90 were 174.83 ± 19.53 nm, 219.23 ± 18.93 nm, and 403.17 ± 32.99 nm, respectively. It was found that a higher ratio of HPBCD (1:8:20) led to OAnf having a larger fiber diameter (Table 1).

#### 3.2.2. Particle Size and Morphology of OAnf Reconstituted in Water

In order to observe the particle shape and particle size of the OAnf (OA:PVP:HPBCD, 1:8:20) dissolved in water, the image of OAnf under the transmission electron microscope (TEM) showed that the oleanolic acid particles were spherical and uniformly dispersed in water (Figure 3). The particle size was also confirmed by a laser particle size analyzer (Table 2). The particle sizes of OA and OAnf were 5079.50 ± 384.87 nm and 302.37 ± 11.91 nm, respectively. The polydispersity indexes (PDI) of OA and OAnf were 1.63 ± 0.21 and 0.32 ± 0.02, respectively. These results indicate that the electrospinning process effectively reduced the particle size of OA with uniform particle distribution and resulted in enhancement of the surface area.

#### 3.2.3. Drug Loading, Encapsulation Efficiency, and Water Solubility of Oleanolic Acid Nanofibers

As shown in Table 3, the loading percentages of OA in different ratios of excipients were 72.36 ± 10.45%, 84.23 ± 3.62%, and 98.19 ± 4.82%, respectively. The results indicated that a higher ratio of HPBCD displayed a better drug loading effect. The encapsulation efficiency of all the formulations was greater than 95%, which indicated that PVPK90 and HPBCD effectively encapsulated OA. In addition, the water solubility of OAnf with different ratios of excipients were 296.14 ± 57.75 µg/mL, 395.87 ± 32.77 µg/mL, and 998.7 ± 58.32 µg/mL, respectively. These results showed that the increase in HPBCD in the formulation dramatically enhanced water solubility of raw OA. By contrast, the water solubility of raw OA could not be determined due to it being below the detection limit (0.01 µg/mL) in the HPLC method. This result indicated that the 1:8:20 OAnf had more than a 1000-fold water solubility improvement when compared with raw OA. Thus, in the following studies, 1:8:20 OAnf was used to determine the biological activity in the PM-induced keratinocytes damage model.

#### 3.2.4. Crystalline Change of Oleanolic Acid and Its Nanofibers

The X-ray diffraction (XRD) patterns of raw OA, excipients, and its nanofibers are shown in Figure 4. Raw OA exhibited multiple high-intensity characteristic diffraction peaks at a scanning angle of 5°–20°, indicating that raw OA was a crystalline compound. On the other hand, the diffraction patterns of PVPK90 and HPBCD have no obvious characteristic diffraction peaks. In addition, for raw OA under electrospinning processing, all the characteristic diffraction peaks of the raw OA completely disappeared, which indicated that the nature of raw OA was transformed from crystalline to amorphous (Figure 4). According to these results, we could conclude that raw oleanolic acid was successfully encapsulated into HPBCD and encapsulated by PVPK90 after the nanofiber process.

#### 3.2.5. Intermolecular Hydrogen Bond Formation between Oleanolic Acid and Excipients

The intermolecular interaction of raw OA and HPBCD with PVPK90 was determined by FTIR spectroscopy, and the results are demonstrated in Figure 5. The FTIR spectrum clearly showed the absorbance of several chemical functional groups of raw OA, including an absorption band at 3463 cm^−1^ (—OH stretch vibration), 1696 cm^−1^ (—C=O stretch vibration), and 1462 cm^−1^ (—CH_2_ stretch vibration) (Figure 5). When OA, PVPK90, and HPBCD were complexed to form nanofibers, the absorbance of these chemical functional groups obviously shifted to a lower absorption. These findings were indicative of intermolecular hydrogen bond interactions between OA and HPBCD with PVPK90. In addition, the present study also used ^1^H NMR to confirm the intermolecular interaction of OA and excipients. The 1H NMR spectrum of raw OA (Figure 6C) showed a carboxyl signal at δ12 ppm (H28), double bound protons at δ5.15 ppm (H12), hydroxy proton signal at δ3.38 ppm (H3), and methyl protons (δ1 ppm). However, the ^1^H NMR spectrum of OA nanofibers showed that carboxyl signal of OA disappeared, and the chemical shifts of double bound, hydroxy, and methyl protons were obviously moved upfield (Figure 6). These results demonstrated the formation of intermolecular hydrogen bonds between OA and excipients, which supported the successful encapsulation of OA by HPBCD and PVPK90.

#### 3.2.6. In Vitro Skin Penetration of Raw Oleanolic Acid and Its Nanofibers

The biological activity of a topical formulation mostly depends on the skin absorption. The skin penetrations of the raw OA and its nanofibers were determined in ex vivo pig skin. As shown in Figure 7, a lower content of OA (<5 µg/cm^2^) was detected in the epidermis and dermis after 1, 2, and 4 h of topical administration, and these results also indicated that raw OA could not penetrate the skin in a time-dependent manner. The result showed that the skin absorption of raw OA was extremely poor. By contrast, the nanofiber formulation dramatically increased the content of OA in the epidermis and dermis with 19.63 µg/cm^2^, 31.56 µg/cm^2^, and 45.27 µg/cm^2^ after 1, 2, and 4 h of topical administration, respectively. These results demonstrated that the OAnf formulation significantly increased skin absorption when compared with the raw OA topical administration (*p* < 0.05).

### 3.3. Oleanolic Acid Nanofibers at Non-Cytotoxic Concentrations Had Better Anti-Pollutant Activity by Improving Antioxidant, Anti-Inflammatory, and Anti-Aging Activity

Similar to the OA dissolved in DMSO, the OAnf dissolved in PBS at 40 µM and 80 µM reduced the viability of HaCaT cells to 14.1% and 5.6%, respectively (Figure 8A). Therefore, 10 µM of OAnf were evaluated for further antioxidant and antipollution activity in order to understand whether OA and its nanofibers have the ability to inhibit the excessive production of ROS caused by PM. Figure 8B shows that OAnf at 10 µM notably reduced PM-induced ROS overproduction. We also calculated the rate of inhibition of ROS production to compare the antioxidant activity between PBS-dissolved OA and OAnf. Ten micromolar OA in PBS resulted in a 28.3% inhibition of ROS production, and OAnf achieved 97.6% inhibition (Figure 8B). These results indicated that OAnf had better antioxidant activity than OA in PBS in PM-induced oxidative stress in keratinocytes. In addition, we also compared the anti-inflammatory activity of PM-induced keratinocytes damage. Pretreatment with raw OA in PBS could not inhibit the PM-induced protein expression of NF-κB and COX-2. In contrast, pretreatment with OAnf in PBS significantly diminished the expression of NF-κB and COX-2 in PM-treated cells (*p* < 0.05). These results supported that OAnf in PBS had better anti-inflammatory activity than raw OA in PBS (Figure 8C). Then, we also compared their anti-skin-aging activity. Pretreatment with OA in PBS had no effects on PM-induced MMP-1 or TIMP-1 alteration. However, OAnf in PBS could reduce the expression of MMP-1 and rescue the expression of TIMP-1 when compared with the PM-induced keratinocytes damage group (*p* < 0.05) (Figure 8D). These findings indicated that OAnf possessed better anti-skin-aging properties than raw OA in PBS. Finally, we analyzed the phosphorylation of ERK, JNK, and p38 to confirm the regulation of MAPKs signaling. Figure 8E showed that the treatment of raw OA in PBS could not downregulate PM-induced phosphorylation of these MAPKs protein. However, pretreatment with OAnf only reduced PM-induced phospho-JNK (p-JNK) expression but had no effect on p-ERK and p-p38 (Figure 8E). The percentage changes of protein expression induced by OA in DMSO, OA in PBS, and OAnf in PBS are summarized at Table 4. OAnf in PBS markedly reversed PM-induced protein alterations, which was barely observed in OA in the PBS group. In addition, the effects of OAnf in PBS were comparable to the equivalent amount of OA in DMSO, which indicated that the electrospinning process increased the water solubility of OA without altering its bioactivities. Accordingly, OAnf effectively inhibited the expressions of inflammatory proteins and skin-aging proteins and downregulated the MAPKs signaling pathway in PM-induced keratinocytes damage.

## 4. Discussion

Recently, monitoring the concentration of particulate matters in the air has become one of the most important indicators used to evaluate the air quality index. The skin is the largest immune organ of human beings, and PM overexposure could damage skin functions. Jin et al. revealed that various types of PM not only stayed in the outside stratum corneum of the epidermis but also penetrated into the stratum spinosum and hair follicles [14]. If exposed to excessive PM for a long time, this will cause skin barrier dysfunction and is associated with many skin diseases, such as atopic dermatitis, psoriasis, acne, and aging [19,20]. Dijkhoff et al. clearly illustrated that PM can trigger exogenous and endogenous ROS formation, resulting in oxidative stress progression, including lipid peroxidation, protein oxidation, mitochondrial dysfunction, DNA damage, inflammation activation, and aging process acceleration [15]. Accordingly, counteracting PM-induced ROS overproduction is a good strategy and the first choice to prevent oxidative stress damage during PM overexposure. Previous studies have also demonstrated that antioxidant treatments, such as diphlorethohydroxycarmalol [21], dieckol [22], *Opuntia humifusa* extract [23], and tart cherry extract [24] can effectively attenuate PM-induced keratinocytes dysfunction. Our results also found that OA in DMSO can reduce PM-induced ROS overproduction and prevent the damage from PM. In addition, NF-κB is a ubiquitous and inducible transcription factor regulating the expression of proinflammatory proteins, such as COX-2, which play a critical role in many skin diseases. Our results mentioned that OA can diminish the PM-induced activation of NF-κB to inhibit the expression of inflammatory protein COX-2 [25]. Moreover, activation of MAPKs signaling can enhance the expression of AP-1 and leads to transcriptional regulation of MMPs [18]. In this study, OA was able to suppress the expression of skin-aging protein MMP-1 and increase the expression of anti-skin-aging protein TIMP-1 to prevent PM-induced aging of keratinocytes. Our results also demonstrated that the OA was effectively inhibited the phosphorylation of JNK. Therefore, OA may inhibit PM-induced skin inflammation and aging by downregulating the ROS/JNK signaling pathway in PM-induced keratinocytes damage.

To the best of our knowledge, the poor water solubility of the active compounds is associated with low bioavailability, which limits their application in medicine, food, and cosmetic industries [26,27]. It is well known that compounds with poor water solubility have several common physicochemical features, such as excessively large particle size, a lower surface area, a lipophilic structure, and a crystalline form [28]. Our results also indicated that raw OA had these physicochemical properties, including a large particle size (5079.50 ± 384.87 nm), a 3–60 μm irregular granular powder with lower surface area (Figure 2C), and an obvious crystalline form. These results indicated that the water solubility of OA was lower than 0.01 μg/mL and could be classified as a practically insoluble active compound according to the water solubility classification of the United States Pharmacopeia (USP) [29]. If these drawbacks cannot be solved, the activities of OA on the skin would be greatly limited. The present study successfully used PVPK90 and HPBCD as carriers using an electrospinning process to prepare OAnf. The water solubility improvement is the major index used to confirm the optimal pharmaceutical formulation, and our results demonstrated that OA:PVPK90:HPBCD at 1:8:20 had the best water solubility. This indicated that OAnf effectively increased the water solubility of raw OA depending on the HPBCD ratio. Similarly, previous studies have also revealed that a higher ratio of cyclodextrin enhances the encapsulation capacity of the formulation and results in a significant increase in the water solubility and biological activity of curcumin [30], resveratrol [12], thymol [31], and difenoconazole [32]. Moreover, the present study also compared the physicochemical properties of raw OA and its nanofibers to elucidate the mechanisms of water solubility improvement of OA. OA nanofibers produced in this study were all filaments with a uniform nano-size. The particle size analysis results also mentioned that OAnf reconstituted in water exhibited nanosized particles with superior distribution homogeneity. These results indicated that OAnf had a greater surface area than raw OA. In addition, the formation of intermolecular hydrogen bonding between active compounds and carriers could also contribute to the improvement of water solubility. The spectrum of FTIR and ^1^H NMR of OAnf demonstrated that raw OA was effectively encapsulated into HPBCD and formed a stabilized nanofiber structure with PVPK90 through forming the intermolecular hydrogen bonding between OA and HPBCD/PVPK90. The crystalline form transformed to an amorphous form of the active compound was also an indicator of water solubility improvement. The XRD pattern of OAnf showed that the crystallin structure of raw OA was transformed to an amorphous structure after the formation of nanofibers. This similar result was also observed in several active compounds loaded in nanofibers [12,30]. Taken together, the nanofiber formulation effectively improved the water solubility of raw OA through improving the physicochemical properties, including a particle size reduction, a surface area increase, hydrogen bonding formation with carriers, and amorphous transformation.

Topical formulations containing antioxidants are effective for delivery to the epidermis and dermis to counteract PM-induced oxidative stress, inflammation, and skin aging [33]. Based on knowledge of skin absorption, the stratum corneum is the rate-limiting factor that limits skin penetration and absorption of active compounds, resulting in a decrease in biological activity [34]. The result from the in vitro skin penetration indicated that OAnf passed through the stratum corneum more easily and quickly than the raw OA and stayed in the epidermis and dermis in large amounts. This result confirmed that OAnf can effectively improve the skin absorption of raw OA. Next, in order to determine whether the OAnf had better antipollution activity than the raw OA, in this study, a PM-induced keratinocytes damage model was used to compare their biological activities. Our results demonstrated that OAnf in PBS had better antipollution effects than raw OA, including reduced ROS overproduction, decreased inflammatory protein expression (COX-2 and NF-κB) and skin-aging protein (MMP-1), increased anti-skin-aging protein expression (TIMP-1), and downregulated JNK phosphorylation. Therefore, OAnf could be a topical formulation as an antioxidant agent for preventing PM-induced keratinocytes damage.

In summary, OAnf improved its physicochemical properties to solve the poor water solubility of raw OA and also significantly improved the skin absorption of raw OA. OAnf had better antioxidant, anti-inflammation, and anti-aging activities in PM-induced keratinocytes damage. Consequently, we suggest that OAnf could be used as a skin care product or as a pharmaceutical formulation to prevent PM-induced skin damage in the future.

## Figures and Tables

**Figure 1 antioxidants-10-01411-f001:**
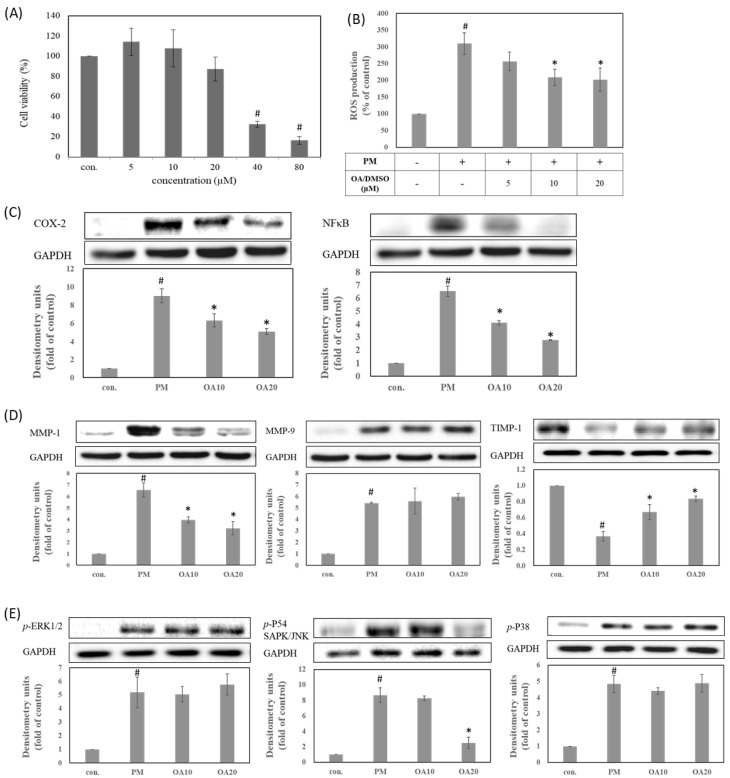
Oleanolic acid inhibits protein expression related to inflammation and aging in PM−induced keratinocytes damage through the ROS/MAPKs pathway. (**A**) Cell viability, (**B**) ROS production, (**C**) COX-2 and NF-κB, (**D**) MMP and TIMP protein expression, and (**E**) phosphorylation of MAPKs. # represents *p* < 0.05 when compared with negative control. * represents *p* < 0.05 when compared with PM-induced control.

**Figure 2 antioxidants-10-01411-f002:**
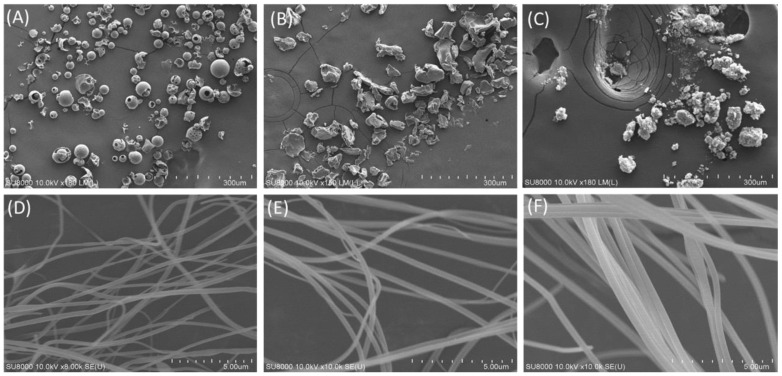
Surface morphology of excipients, oleanolic acid (OA), and its nanofibers under scanning electron microscope (SEM). (**A**) HBPCD, (**B**) PVPK90, (**C**) OA, and the different ratios of oleanolic acid nanofibers (OA:PVP:HPBCD, *w*/*w*/*w*) (**D**) 1:8:5, (**E**) 1:8:10, (**F**) 1:8:20.

**Figure 3 antioxidants-10-01411-f003:**
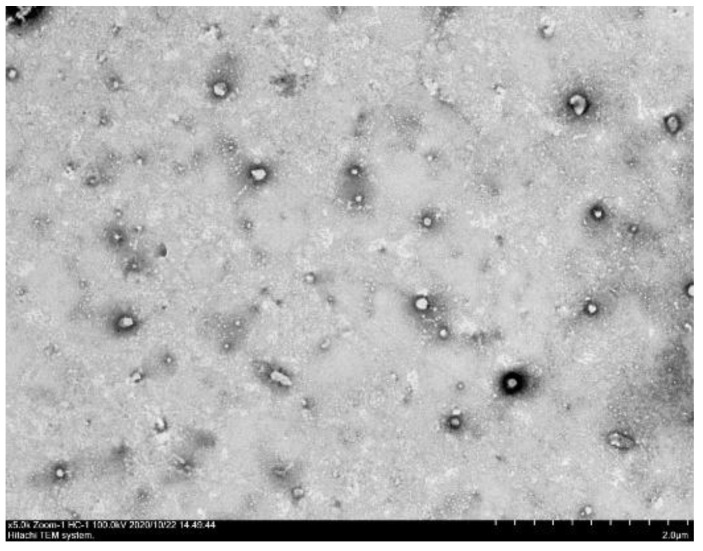
Morphology of the nanofiber of oleanolic acid (OAnf 1:8:20) under transmission electron microscope (TEM).

**Figure 4 antioxidants-10-01411-f004:**
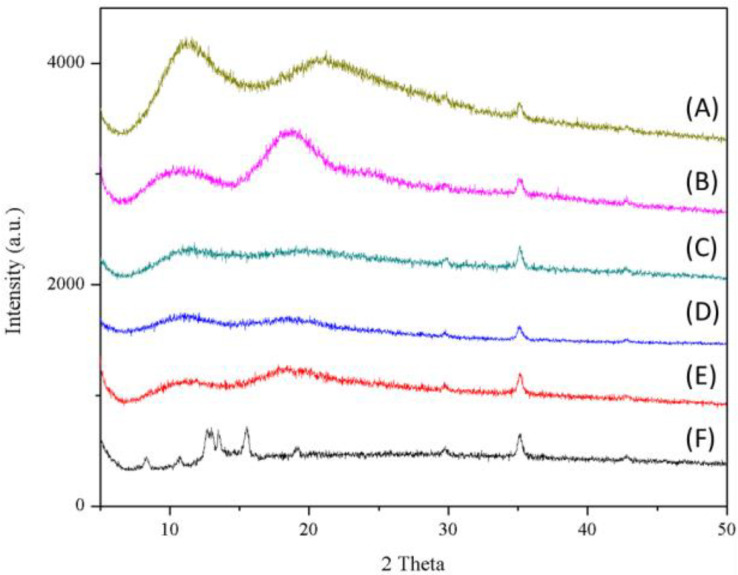
X-ray diffractometer (XRD) spectra of raw OA, OAnf with different ratios, and excipients. (**A**) PVPK90, (**B**) HPBCD, the different ratio of OAnf (OA:PVP:HPBCD, *w*/*w*/*w*) (**C**) 1:8:5, (**D**) 1:8:10, (**E**) 1:8:20, and (**F**) oleanolic acid.

**Figure 5 antioxidants-10-01411-f005:**
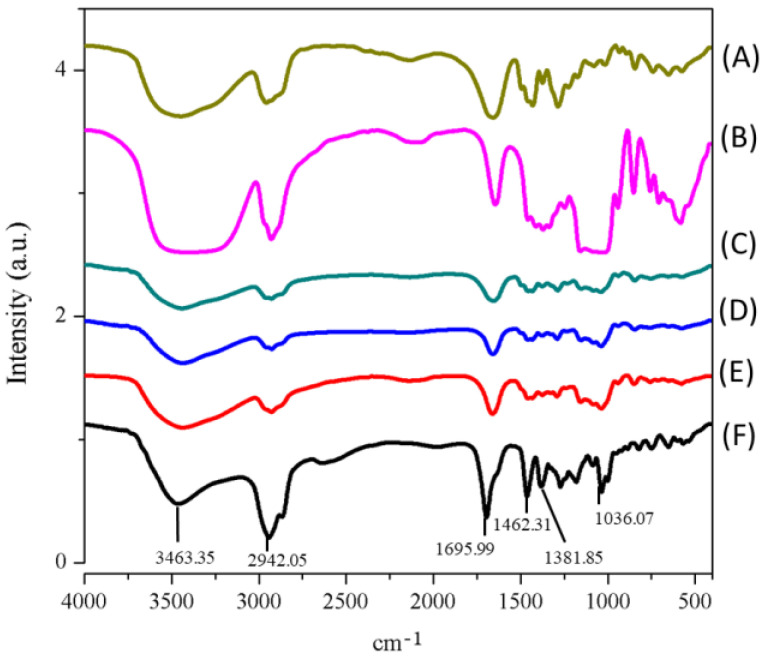
Fourier transform infrared (FTIR) spectra of OA and OAnf with different ratios and excipients. (**A**) PVPK90, (**B**) HPBCD, the different ratios of OAnf (OA: PVP: HPBCD, *w*/*w*/*w*) (**C**) 1:8:5, (**D**) 1:8:10, (**E**) 1:8:20, and (**F**) oleanolic acid.

**Figure 6 antioxidants-10-01411-f006:**
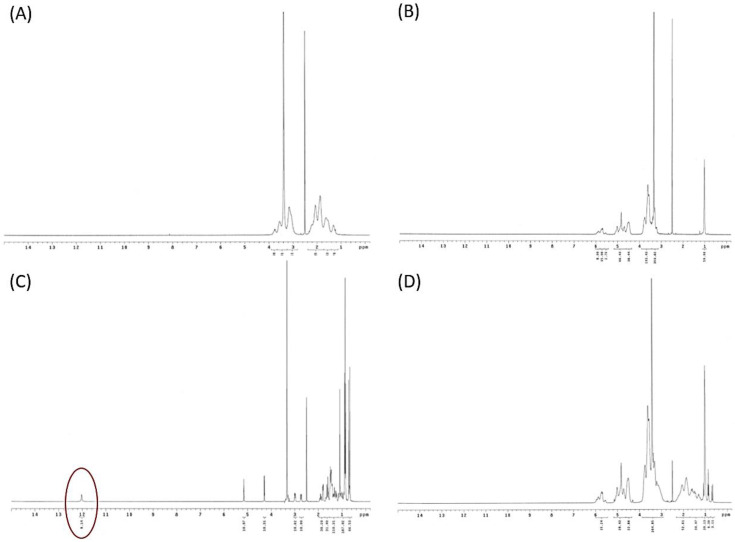
Nuclear Magnetic Resonance (NMR) spectra of OA and OAnf with different ratios and excipients. (**A**) HPBCD, (**B**) PVPK90, (**C**) oleanolic acid, and (**D**) OAnf (1:8:20).

**Figure 7 antioxidants-10-01411-f007:**
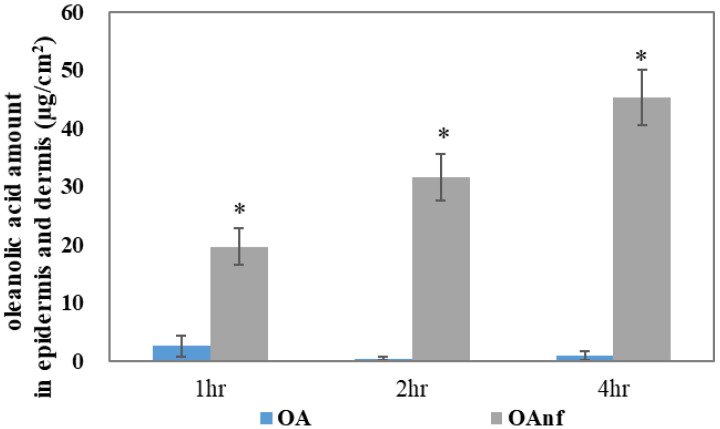
In vitro skin penetration of raw OA and OAnf in ex vivo pig skin. The amount of OA in the epidermis and dermis at various time points was determined. Results are shown as mean ± SD (*n* = 5). * *p* < 0.05 significance with raw OA.

**Figure 8 antioxidants-10-01411-f008:**
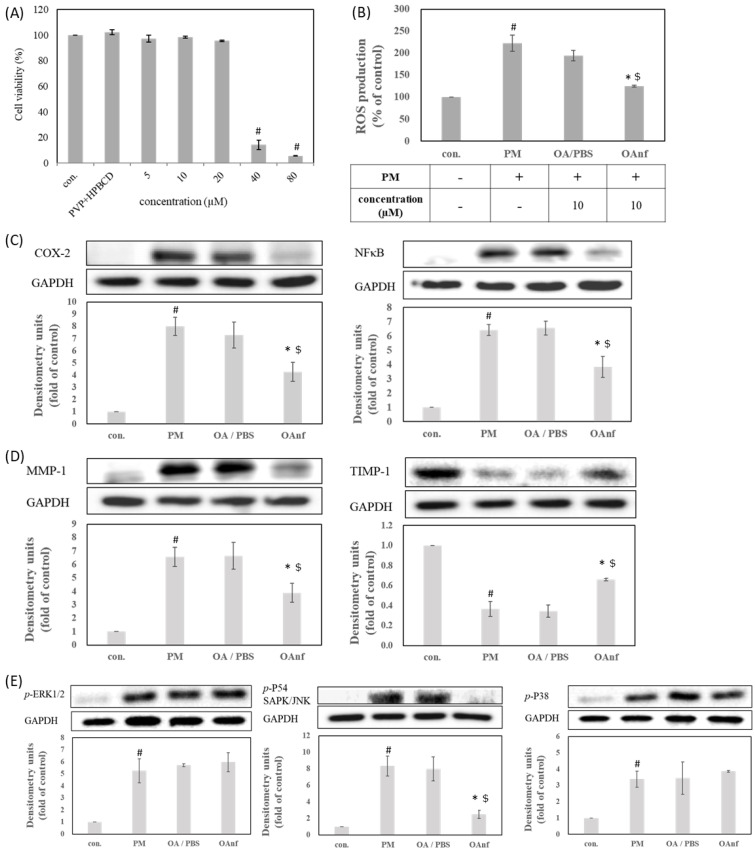
Oleanolic acid nanofiber inhibits the protein expression of inflammation and aging in PM−induced keratinocytes damage through the ROS/MAPKs pathway. (**A**) Cell viability, (**B**) ROS production, (**C**) COX-2 and NF-κb, (**D**) MMPs and TIMP-1 protein expression, (**E**) phosphorylation of MAPKs. A *p*-value < 0.05 was interpreted as statistically significant, # refers to PM treatment versus negative control, * refers to PM treatment versus OAnf, and $ refers to OA/PBS versus OAnf.

**Table 1 antioxidants-10-01411-t001:** Diameter of oleanolic acid and its nanofibers under scanning electron microscope (SEM).

OA:PVP:HPBCD (*w*/*w*/*w*)	Diameter (nm)
1:8:5	174.83 ± 19.53
1:8:10	219.23 ± 18.93
1:8:20	403.17 ± 32.99

**Table 2 antioxidants-10-01411-t002:** Diameters of oleanolic acid and its nanofibers detected by laser particle size analyzer. Results are shown as mean ± SD of three independent experiments.

	Particle Size (nm)	Polydispersity Index (PDI)
pure oleanolic acid	5079.50 ± 384.87	1.63 ± 0.21
OAnf	302.37 ± 11.91	0.32 ± 0.02

**Table 3 antioxidants-10-01411-t003:** Drug loading, encapsulation efficiency, and water solubility of different formulations of OAnf. Results are shown as mean ± SD of three independent experiments.

Ratio (OA:PVP:HPBCD)	Drug Loading (%)	Solubility (µg/mL)	Encapsulation Efficiency(%)
raw oleanolic acid	-	˂LOD *	-
1:8:5	72.36 ± 10.45	296.14 ± 57.75	96.92 ± 3.15
1:8:10	84.23 ± 3.62	395.87 ± 32.77	>99
1:8:20	98.19 ± 4.82	998.7 ± 58.32	>99

* LOD: Limit of detection (<0.01 μg/mL).

**Table 4 antioxidants-10-01411-t004:** The percentage change of protein expression from OA and OAnf in PM-induced HaCaT cell damage.

Protein	Percentage Change (%) ^a^		
	OA/DMSO	OA/PBS	OAnf/PBS
COX-2	49.01 ± 6.34	19.76 ± 12.03	54.37 ± 9.2
NF-κB	68.01 ± 1.87	NE ^b^	51.7 ± 6.33
MMP-1	60.18 ± 8.02	NE	45.14 ± 8.75
TIMP-1	75.66 ± 3.13	NE	61.57 ± 3.02
JNK	86.54 ± 6.13	9.13 ± 16.43	82.31 ± 4.42

^a^ Percentage change was calculated from the quantitative results of Figure 1 and Figure 8 by the following equation: |Fold change^PM^ –Fold change^OA^|/|Fold change^PM^ –Fold change^Con^| × 100%. Values are expressed as mean ± SD. ^b^ NE: no effect.

## Data Availability

The data presented in this study are available in article.

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
