# Peer review of "Oleanolic Acid Nanofibers Attenuated Particulate Matter-Induced Oxidative Stress in Keratinocytes"

_antioxidants, 2021, doi:10.3390/antiox10091411_

Round 1

Reviewer 1 Report

  Thank you very much for submitting your Manuscript ID antioxidants-1325161 “Oleanolic acid nanofiber displayed an antioxidant to attenuate particulate matter-induced oxidative stress in keratinocytes” for the antioxidants.  I think a good idea using an oleanolic acid nanofiber is able to use the lotion products and its displayed by antioxidant is useful to evaluate the effect to keratinocytes damaged by PM.  Before accepting your manuscript, I think the following issues are revised.

1)     Abstract, line 2 from the bottom, keratinocyte injury to keratinocyte damage, because this is in vitro study and not to be used the term of injury.  that same as below in the manuscript.

2)     The commercial information of fresh pig skin is shared at Materials.

3)   Materials and Methods, 17 small items  are too much.  The classification in items should be done.

4)   I guess the significant effect of OA at 10 µM, DMSO in Fig. 1, however, no effect of OA at 10 µM, PBS in Fig. 8 and similar level between OA and OAnf.  These results are summarized at Table instead of Figure 8. It is difficult for users to recognize the effect of OAnf compared with Fig.1 and Fig.8. 

   I think this manuscript is accepted for this journal after minor revision.    Thank you.

Author Response

Response to Reviewer 1

Reviewer 1

Thank you very much for submitting your Manuscript ID antioxidants-1325161 “Oleanolic acid nanofiber displayed an antioxidant to attenuate particulate matter-induced oxidative stress in keratinocytes” for the antioxidants.  I think a good idea using an oleanolic acid nanofiber is able to use the lotion products and its displayed by antioxidant is useful to evaluate the effect to keratinocytes damaged by PM.  Before accepting your manuscript, I think the following issues are revised.

1)     Abstract, line 2 from the bottom, keratinocyte injury to keratinocyte damage, because this is in vitro study and not to be used the term of injury.  that same as below in the manuscript.

Reply: Thanks for your suggestion. ‘Keratinocyte injury’ was replaced by ‘keratinocyte damage’ throughout the revised manuscript.

2)     The commercial information of fresh pig skin is shared at Materials.

Reply: Fresh flank skin from pig was obtained from a local butcher in the market and refrigerated during the experiment period. (Page 6, Line 242)

3)   Materials and Methods, 17 small items are too much.  The classification in items should be done.

Reply: Thanks for your suggestion. We have reduced 17 items to 13 items. 

4)   I guess the significant effect of OA at 10 µM, DMSO in Fig. 1, however, no effect of OA at 10 µM, PBS in Fig. 8 and similar level between OA and OAnf. These results are summarized at Table instead of Figure 8. It is difficult for users to recognize the effect of OAnf compared with Fig.1 and Fig.8. 

Reply: Thanks for your comment. We added a new table to compare the percentage changes of protein expression induced by OA in DMSO, OA in PBS, and OAnf in PBS (Table 4, Section 3.3, page 12-14). Our results showed that OAnf in PBS markedly reversed PM-induced protein alterations, which was barely observed in OA in the PBS group. In addition, the effects of OAnf in PBS were comparable to an equivalent amount of OA in DMSO, which indicated that the electrospinning process increased the water solubility of OA without altering its bioactivities.

Reviewer 2 Report

General comments

The article needs English editing, because many sentences have no sense and are misleading.

Authors made a lot of experiments, but the article is chaotically written. It must be improved. Methodology must be written in a logic way. What was tested and why? PM? OA? OA and PM together?

The Authors must precise the aim of the study – general and detailed, because in the current form it is confusing.

Detailed comments

The title is not grammatically correct: …antioxidant… what? Properties? Activity?

Abstract:

The sentences “Our result demonstrated that OA can reduce the content of reactive oxygen species, the protein expression of inflammatory, aging, and SPAK/JNK phosphorylation to attenuate PM-induced oxidative stress in keratinocytes.”

“Moreover, OAnf displayed better antioxidant, anti-inflammatory, and antiaging in PM-induced keratinocyte injury via ROS/COX-2/JNK pathways.”

are not grammatically correct.

Introduction

Authors should give some information about and justify the choice of polyvinyl pyrrolidone (PVPK90) and 2-hydroxypropyl-β-cyclodextrin (HPBCD) in their study.

Give research hypotheses.

Methods
Paragraph 2.2.

HaCaT cells: what passage was used? Why this cell line was used? Justify.

The sentence (102-104): “In addition, the fetal bovine serum must be removed from the medium used for prepare of drugs to prevent the cells from affecting the absorption of the drugs.” Must be rebuilt in grammar and style.

Give more detail about cell culture.

What authors wanted to demonstrate with the experiment?

What is cell safety assay… - MTT? Give detailed description how the cells were treated. Why 24 hours? Why MTT?

What is the standard error of the method in your laboratory?

Line 105-106: “The culture medium was removed and the cells were treated with different concentrations of each sample in treatment medium.” With different concentration of what? PM? OA? How? What concentrations? How many? How did you chose them – on the basis of what?

Line 107-108: how did you dissolve formazan crystals? Did you measure absorbance of/with MTT?

Paragraph 2.4.

How did you apply PM on the surface of cells? Was the PM sterile? How did you sterilize it? I understand that PM produce suspension of particles, which can adhere to cells. Did these particles interfere with the fluorescence measurement? Washing cells 2 times can be not enough… Give photos.

There should be a positive control. This data is unreliable without positive control.

Paragraph 3.2.6.

Give photographs.

Author Response

Response to Reviewer 2

Reviewer 2

General comments

The article needs English editing, because many sentences have no sense and are misleading.

Reply: Thanks for your comment. The revised manuscript has been edited by the English editing service of MDPI.

Authors made a lot of experiments, but the article is chaotically written. It must be improved. Methodology must be written in a logic way. What was tested and why? PM? OA? OA and PM together?

Reply: Thanks for your suggestion. We have rewritten the Methods section.

The Authors must precise the aim of the study – general and detailed, because in the current form it is confusing.

Reply: Thanks for your suggestion. We have rewritten the Introduction section.

Detailed comments

The title is not grammatically correct: …antioxidant… what? Properties? Activity?

Reply: Thanks for your comment. We have modified the title as follows: “Oleanolic acid nanofibers attenuated particulate matter-induced oxidative stress in keratinocytes”

Abstract:

The sentences “Our result demonstrated that OA can reduce the content of reactive oxygen species, the protein expression of inflammatory, aging, and SPAK/JNK phosphorylation to attenuate PM-induced oxidative stress in keratinocytes.”

“Moreover, OAnf displayed better antioxidant, anti-inflammatory, and antiaging in PM-induced keratinocyte injury via ROS/COX-2/JNK pathways.”

are not grammatically correct.

 Reply: Thanks for your comment. We have rewritten the Abstract, and the manuscript was edited with the English editing service of MDPI. 

Introduction

Authors should give some information about and justify the choice of polyvinyl pyrrolidone (PVPK90) and 2-hydroxypropyl-β-cyclodextrin (HPBCD) in their study.

Give research hypotheses.

Reply: We have added following information about polyvinyl pyrrolidone (PVPK90) and 2-hydroxypropyl-β-cyclodextrin (HPBCD) in the Introduction section (Page 2, Line 76-81):

Polyvinyl pyrrolidone (PVPK90) and 2-hydroxypropyl-β-cyclodextrin (HPBCD) are the FDA-approved compounds for solubilizing and delivering hydrophobic active pharmaceutical ingredients in humans. Previous studies showed that nanofibers prepared with HPBCD and PVPK90 significantly improved the water solubility and skin penetration of resveratrol [12] and plai oil [13].

References:

[12] Lin YC, Hu SC, Huang PH, Lin TC, Yen FL. Electrospun Resveratrol-Loaded Polyvinylpyrrolidone/Cyclodextrin Nanofibers and Their Biomedical Applications. Pharmaceutics. 2020 Jun 13;12(6):552.

[13] Tonglairoum P, Chuchote T, Ngawhirunpat T, Rojanarata T, Opanasopit P. Encapsulation of plai oil/2-hydroxypropyl-beta-cyclodextrin inclusion complexes in polyvinylpyrrolidone (PVP) electrospun nanofibers for topical application. Pharm Dev Technol. 2014 Jun;19(4):430-7.

Methods
Paragraph 2.2.

HaCaT cells: what passage was used? Why this cell line was used? Justify.

Reply: Thanks for pointing this out. HaCaT cells from passage 5 to 10 were used in this study. HaCaT cells are a keratinocyte cell line commonly used to evaluate dermatological effects of active ingredients in vitro.

The sentence (102-104): “In addition, the fetal bovine serum must be removed from the medium used for prepare of drugs to prevent the cells from affecting the absorption of the drugs.” Must be rebuilt in grammar and style.

Give more detail about cell culture.

Reply: Thanks for your comment. We have rewritten Section 2.2, and the manuscript was edited with the English editing service of MDPI.

What authors wanted to demonstrate with the experiment?

Reply: Thanks for pointing this out. The cell viability assay was used to determine the proper concentration range of the active ingredient for further biological activity evaluation.

What is cell safety assay… - MTT? Give detailed description how the cells were treated. Why 24 hours? Why MTT?

Reply: Thanks for your comment. ‘Cell safety assay’ was replaced by ‘Cell viability assay’. We also rewrote Section 2.2. MTT is one of the most common tests for determining cell viability. Regarding the treatment duration, we used 24 hours because cosmetic products do not usually stay on the skin’s surface over 24 hours.

What is the standard error of the method in your laboratory?

Reply: The MTT assay worked quite well in our laboratory. Mostly, the coefficient of variation (CV) (or relative standard deviation (RSD)) of replicates in an experiment was around 5%. The results of the cell viability assay were calculated based on three independent experiments. In each experiment, three replicates were performed for each testing group. 

Line 105-106: “The culture medium was removed and the cells were treated with different concentrations of each sample in treatment medium.” With different concentration of what? PM? OA? How? What concentrations? How many? How did you chose them – on the basis of what?

Reply: Thanks for your comment. We have rewritten this paragraph. These lines were amended as follows:

The culture medium was then removed, and the cells were treated with different concentrations of OA and OAnf ranging from 5 to 80 μM in serum-free DMEM for 24 hours.

The concentrations of OA for the cell viability assay were referenced from the following reports:

George VC, Kumar DR, Suresh PK, Kumar RA. Apoptosis-induced cell death due to oleanolic acid in HaCaT keratinocyte cells--a proof-of-principle approach for chemopreventive drug development. Asian Pac J Cancer Prev. 2012;13(5):2015-20.

Kuonen R, Weissenstein U, Urech K, Kunz M, Hostanska K, Estko M, Heusser P, Baumgartner S. Effects of Lipophilic Extract of Viscum album L. and Oleanolic Acid on Migratory Activity of NIH/3T3 Fibroblasts and on HaCat Keratinocytes. Evid Based Complement Alternat Med. 2013;2013:718105.

Line 107-108: how did you dissolve formazan crystals? Did you measure absorbance of/with MTT?

Reply: Thanks for your comment. We have rewritten this paragraph. The relevant lines were amended as follows:

At the time of assay, the treatment medium was removed, and 150 μL of 0.5 mg/mL MTT solution was added into each well. After 3 hours of incubation, the MTT solution was removed, and the purple formazan crystals of each well were dissolved in 100 μL of DMSO. The absorbance at 550 nm of each well was then measured using a microplate spectro-photometer.

Paragraph 2.4.

How did you apply PM on the surface of cells? Was the PM sterile? How did you sterilize it? I understand that PM produce suspension of particles, which can adhere to cells. Did these particles interfere with the fluorescence measurement? Washing cells 2 times can be not enough… Give photos.

Reply: Thanks for your comment. The PM used in this study is Standard Reference Material, SRM® 1649b, which is not sterile. We suspended PM in sterile PBS before use, and we did not observe any sign of microbial contamination throughout the experiment. Although PM is difficult to remove, even with more than 5 washes, we did not observe fluorescence emitted from the aggregated particles of the PM in our preliminary experiments. As shown in the figure below, HaCaT cells were treated with PM and stained with DCFDA for the detection of ROS level. Emissions of DCFDA were detected with an FITC filter set. The black particles with various sizes are PMs, which can be observed in both bright field and FITC channels. There was no fluorescence emitted from these particles.

There should be a positive control. This data is unreliable without positive control.

Reply: Thanks for your suggestion. N-acetyl cysteine (NAC) is a well-known hydrophilic antioxidant compound and is usually used as a positive control to reduce ROS levels. Piao et al. demonstrated that 1 mM (163 μg/mL) of NAC can attenuate 50 μg/mL PM-induced ROS overproduction in HaCaT cells. We have tested the effects of 1 mM, 2 mM, and 4 mM of NAC in PM-treated HaCaT cells; however, we did not observe a reduction of ROS production in any of these groups. One possible explanation is the concentration of PM used in our study (50 μg/cm2 ~= 160 μg/mL) was around 3-fold higher than that in Piao’s study. 

Piao, M.J., Ahn, M.J., Kang, K.A. et al. Particulate matter 2.5 damages skin cells by inducing oxidative stress, subcellular organelle dysfunction, and apoptosis. Arch Toxicol 92, 2077–2091 (2018). https://doi.org/10.1007/s00204-018-2197-9

Paragraph 3.2.6.

Give photographs.

Reply: Figure 7 was already on page 11 in the manuscript. 

Round 2

Reviewer 2 Report

I have no more comments. Authors answered all questions and improved their manuscript.